



# Land- to lake-terminating transition triggers dynamic thinning of a Bhutanese glacier

Yota Sato[1], Koji Fujita[1], Hiroshi Inoue[2], Akiko Sakai[1] and Karma[3]

[1] Graduate School of Environmental Studies, Nagoya University, Nagoya, Japan
[2] National Research Institute for Earth Science and Disaster Resilience (NIED), Tsukuba, Japan
[3] National Centre for Hydrology and Meteorology (NCHM), Thimphu, Bhutan

*Correspondence to*: Yota Sato (yota.sato@nagoya-u.jp)

**Abstract.** There have been rapid increases in both the number and expansion of the proglacial lakes across High Mountain Asia. However, the relationship between proglacial lakes and glacier dynamics remains unclear in the Himalayan region.
Here we present the surface elevation, flow velocity changes, and proglacial lake expansion of Thorthormi and Lugge glaciers in the Lunana region, Bhutanese Himalaya, during the 2000–2018 period using photogrammetry and GPS survey data. The lake expansion and surface lowering rates, and flow velocity field of Lugge Glacier, a lake-terminating glacier, have remained approximately constant since 2000. Conversely, there has been accelerated proglacial lake expansion and a two-fold increase in the thinning rate of Thorthormi Glacier since 2011, as well as a considerable speed-up in the flow
velocity field (>150 m a$^{-1}$). We reveal that the lake formation and transition of Thorthormi Glacier from a land- to lake-terminating glacier have triggered glacier speed-up and rapid thinning via a positive (compressive) to negative (extensional) change in the emergence velocities. This study provides the first evidence of dynamic glacier changes that are associated with proglacial lake formation across the Himalayan region.

## 1 Introduction

A recent deglacial trend has been reported for numerous glaciers across High Mountain Asia (HMA; e.g., Brun et al., 2019; Maurer et al., 2019; Shean et al., 2020), with these glaciers exhibiting spatially heterogeneous thinning patterns (Bolch et al., 2012; Kääb et al., 2012; Brun et al., 2017). There has been a rapid increase in both the number and expansion of the proglacial lakes across HMA owing to this deglacial trend (Zhang et al., 2015; Nie et al., 2017; Shugar et al., 2020), which has been particularly pronounced across the eastern Himalayas (e.g., Gardelle et al., 2011; Chen et al., 2021). Proglacial
lakes form via the coalescence of supraglacial lakes near the glacier terminus (Quincey et al., 2007); their formation suggests the final phase of retreat for these contracting glaciers (Sakai and Fujita, 2010; Benn et al., 2012). The increasing number and evolution of proglacial lakes have led to a rise in the hazardous potential of glacial lake outburst floods (GLOFs). GLOF hazards can be triggered by either unstable terminal moraines or snow/rock avalanches (e.g., Fujita et al., 2008; Westoby et al., 2014), and can cause significant damage to hydropower stations, bridges and buildings that exist downstream of
proglacial lakes (Richardson and Reynolds, 2000).



Proglacial lake formation accelerates glacier mass loss via thermal undercutting and calving at the glacier terminus (e.g., Benn et al., 2007; Sakai et al., 2009). Previous studies have analysed the interaction between proglacial lakes and glacier dynamics using in-situ measurements and remote-sensing methods across HMA (e.g., King et al., 2018; Haritashya et al., 2018; Wei et al., 2021). Recent high-resolution satellite and aerial photogrammetry techniques have led to improved glacier

and proglacial lake studies. For example, Watson et al. (2020) acquired in situ measurements and unmanned aerial vehicle (UAV) photogrammetry across Tsulagi Glacier in the Nepal Himalaya, and estimated the calving volume at the terminus based on iceberg size. Furthermore, previous studies have also reported the retreat and thinning of lake-terminating glaciers in their catchments to a broad regional scale (e.g., Song et al., 2017; Zhang et al., 2019; Maurer et al., 2019). King et al. (2019) reported that the mass loss of lake-terminating glaciers was greater than that of land-terminating glaciers across broad

Himalayan regions, with an observed increase in mass loss after 2000. Proglacial lakes can enhance glacier retreat and thinning. Lake- and land-terminating glaciers also exhibit different fluctuation patterns, even if they are located near each other and/or exist under similar climatic conditions (Liu et al., 2020). Therefore, advancing our understanding of lake-terminating glacier fluctuations is essential for making robust future predictions of the HMA glacier response.

Numerous proglacial lakes have an exceptionally high-risk potential for GLOFs throughout the Bhutan Himalaya (e.g.,

Fujita et al., 2013; Zheng et al., 2021), and lake expansion appears to continue unabated (Ageta et al., 2000; Komori, 2008). Previous lake-terminating glacier studies have been conducted across the Bhutan Himalaya using either in situ measurements or satellite remote-sensing methods to assess their dynamics and evolutions (e.g., Suzuki et al., 2007; Fujita et al., 2008). Tsutaki et al. (2019) revealed contrasting fluctuations between two neighbouring glaciers in the Lunana region using in situ GPS measurements, satellite remote-sensing data and numerical modelling. They reported a greater thinning rate along lake-

terminating Lugge Glacier than along land-terminating Thorthormi Glacier during the 2004–2011 period, which was attributed to their contrasting terminus conditions. They also projected that the thinning rate and flow speed of Thorthormi Glacier could be accelerated if the current land terminus changed to a lake terminus. The terminus of Thorthormi Glacier is now detached from the terminal moraine and has evolved into a lake-terminating glacier. The associated changes in glacier dynamics due to proglacial lake formation have been studied worldwide (e.g., Boyce et al., 2007; Tsutaki et al., 2013);

however, no such study has been undertaken in the Himalayan region to date. Therefore, this study aims to (1) update the fluctuations of two glaciers that have been affected by proglacial lakes in the Bhutan Himalaya, and (2) evaluate the changes in glacier dynamics associated with the transition from land- to lake-terminating conditions. We analysed past in situ measurements and satellite/airborne remote-sensing datasets to achieve this goal.

## 2 Study Site

Our target glaciers, Thorthormi and Lugge, are located in the Lunana region of northern Bhutan (Fig. 1; 28.06° N, 90.18° E). Thorthormi Glacier covers 11.6 km$^2$ based on the GAMDAM glacier inventory (Nuimura et al., 2015; Sakai, 2019), with the 2017 terminus position and its elevation range spanning 4,400–6,900 m above sea level (a.s.l.). Thorthormi Glacier had



contacted the terminal moraine before 2011, and later detached from the terminal moraine and transitioned into a lake-terminating glacier (Tsutaki et al., 2019). Lugge Glacier is located to the east of Thorthormi Glacier and covers an area of 10.0 km$^2$, with its elevation range spanning 4,500–6,900 m a.s.l. Its proglacial lake (Lugge Tsho) has expanded since the 1960s (Komori, 2008), with a maximum lake depth of 126 m reported in 2002 (Yamada et al., 2004). Lugge Glacial Lake caused an outburst flood in October 1994, and damaged the downstream areas (Fujita et al., 2008; Maurer et al., 2020). Both glaciers are debris-covered, and have been reported to be experiencing long-term mass-loss and thinning trends (Bajracharya et al., 2014; Maurer et al., 2016).

## 3 Observations and analysis methods

### 3.1 DGPS and aerial photogrammetry survey

We used the global positioning system (GPS) dataset of Tsutaki et al. (2019), who conducted a kinematic survey with a differential GPS (DGPS; GEM-1, GNSS Technologies) across the on- and off-glacier terrains during the 19–22 September 2011 field campaign. The base station for this survey was installed to the west of Thorthormi Glacial Lake (Fig. 1a). These GPS data points were used to validate the satellite/photogrammetry digital elevation models (DEMs) and compute the surface elevation changes (Sects. 3.2 and 3.3).

We conducted a helicopter photogrammetry survey on 24 March 2018. Four action cameras (GoPro HERO5 Black) were attached to the skids of a helicopter, and acquired 4000 × 3000 pixel images in 1-s shooting mode. The shutter speed, focal length and ISO were fixed at 1/1250 s, 28.3 mm and 100, respectively. We obtained ~7500 photos in total and cropped each photograph by preserving the central 2500 × 2500 pixel area to eliminate the "fisheye effect" of the GoPro camera lens (Girod et al., 2017). We finally employed 3560 images based on the image quality and glacier coverage. These images were processed in Agisoft Metashape Professional Edition 1.7.1 (Agisoft LLC), and the sky view was masked for the terrain data processing.

### 3.2 Terrain data processing

We extracted ground control points (GCPs) from a Pléiades panchromatic orthoimage (0.5-m resolution), which was acquired on 7 November 2017 (Berthier et al., 2014), and its DEM (2.0-m resolution) for the photogrammetry terrain data processing. We first generated a GPS-derived-DEM (GPS-DEM) to assess the vertical accuracy of the Pléiades-derived-DEM (PL-DEM). The 2011 GPS data points (UTM Zone 46N, WGS84) were interpolated using the inverse distance weighted method, and then exported to the same grid size as the PL-DEM in ArcGIS (Tshering and Fujita, 2016; Sato et al., 2021). We employed the standard deviation (SD; σ) of the elevation difference between the PL- and GPS-DEMs on the off-glacier stable terrain as the vertical accuracy of the PL-DEM. We did not use the grid cells with steep slopes (>30°; Fujita et al., 2008; Nuimura et al., 2012). We then eliminated the validation points that were greater than ±3σ from the mean elevation difference as extreme outliers (Mertes et al., 2017). Berthier et al. (2014) reported that the vertical accuracy of the PL-DEM



was improved by shifting the DEM horizontally. We therefore shifted the PL-DEM by ±2 pixels (±4 m) in the northing and

easting directions, computed the elevation difference against the GPS-DEM and confirmed that there was no improvement in the vertical accuracy. Finally, the PL-DEM vertical bias (mean elevation difference: MED) was assessed for 12,009 grid cells, yielding a mean value of 0.26 ± 3.86 m (MED ± SD). We extracted the GCP coordinates from the orthoimage and bias-corrected PL-DEM. Specific topographic features (e.g., boulders, river bending points and dense vegetation spots) on the stable ground were used as GCPs for the photogrammetry terrain data processing.

We used the Structure from Motion (SfM) software in Agisoft Metashape to generate orthoimages and a DEM from the helicopter photogrammetry data (hereafter HP-ortho and HP-DEM, respectively). We overlaid the 77 GCPs that were extracted from the Pléiades products onto the helicopter photogrammetry images (Fig. 1b), and generated both the HP-ortho and HP-DEM at a 0.5-m resolution (Fig. 1a and b). We employed the same approach used in the PL-DEM evaluation to evaluate the vertical bias and accuracy of the HP-DEM by re-generating a 0.5-m resolution GPS-DEM. The vertical accuracy

of the HP-DEM was 0.25 ± 3.70 m (N = 25,474 GPS-DEM grid cells; Fig. S1a); we also applied an elevation change correction (Sect. 3.3) to correct for the vertical bias of the HP-DEM.

## 3.3 Changes in the glaciers and glacial lakes

We calculated the surface elevation change rates (dh/dt) by comparing the 2011 GPS-DEM and 2018 HP-DEM (both at 0.5-m resolution). We used 9,491 and 15,604 grid cells to calculate dh/dt for Thorthormi and Lugge glaciers, respectively (red

tracks in Fig. 1b). We then compared our results with previous elevation change studies. Tsutaki et al. (2019) calculated dh/dt from the overlapping 2004 and 2011 DGPS data; they also computed the spatial distribution of dh/dt for the same period using ASTER-derived DEMs. We also employed the long-term dh/dt data from Brun et al. (2017) and Maurer et al. (2019) to assess the thinning trends of these two glaciers. Brun et al. (2017) computed dh/dt for the 2000–2016 period over the Hindu-Kush-Himalaya region using ASTER-based DEMs, and Maurer et al. (2019) calculated dh/dt for the 1975–2000

and 2000–2015 periods using satellite-based DEMs. These datasets (hereafter RS-based dh/dt) are provided as 30-m resolution raster data. We extracted the dh/dt data from our DEMs at the same positions for a comparative analysis.

We computed the surface velocity field using the ImGRAFT (Image GeoRectification and Feature Tracking) open-source feature tracking toolbox in MATLAB (Messerli and Grinsted, 2015). The feature-tracking algorithm in the toolbox (Templatematch) identifies the displacement patterns of the glacier surface features and computes their magnitude from a

pair of images. We selected a Sentinel-2 image pair that was acquired on 16 November 2016 and 11 November 2017 (post-monsoon seasons). We chose a $75 \times 75$ search window size ($750 \times 750$ m) after visual trial and error to compute the surface feature displacements and calculate the annual flow velocity. We set the correlation value for image matching ($r > 0.6$) and signal to noise ratio (SNR > 0.7) and eliminated the low-quality pixels, all of which served as a confidence level threshold for successful image matching. We then estimated the uncertainty of the glacier surface velocity and corrected systematic

error by checking the stable-ground (off-glacier) displacement (e.g., Liu et al., 2020). We calculated the stable-ground





surface displacement (slopes < 20°; Quincey et al., 2009), and set the corrected median values of $V_x$ (east/west component) and $V_y$ (north/south component) to zero. The flow speed V (m a$^{-1}$) is calculated as:

$$V = \sqrt{V_x^2 + V_y^2}. \tag{1}$$


The mean and median stable-ground V values were 2.2 and 1.6 m a$^{-1}$, respectively, after the displacement correction. The velocity profiles were extracted along the glacier central flowlines every 10 m, and the pixel values where the flow directions differed by >90° from the flowlines were eliminated. We also eliminated the velocity data along the upper section of Lugge Glacier (>5100 m elevation) because of its heavy snow cover, which can cause incorrect image matching via feature tracking

(e.g., Nuimura et al., 2017). We extracted the surface velocity from the 2017 ITS_LIVE velocity product, which covered the entire HMA region and possessed a 240-m spatial resolution (Gardner et al., 2019). We also employed the velocity data produced by Tsutaki et al. (2019), which were calculated annually from the ASTER-derived optical satellite images at 15-m resolution during the 2002–2011 period.

We delineated the glacial lake area from Landsat 7 and 8 (ETM+/OLI) images with false-colour image composites that were acquired between November 2012 and November 2018 (30-m resolution). We then combined the proglacial lake

polygons before 2012 (Tsutaki et al., 2019), and traced the annual lake area changes for the entire 18-year study period. The total lake area uncertainties were estimated to be ±0.14 and ±0.08 km$^2$ for Thorthormi and Lugge glaciers, respectively, depending on the user-induced error and satellite image resolution (Paul et al., 2013). We removed the DEM and velocity data where the glacier surface turned into the lake surface in successive images. We compared the recent lake expansion

rates with a glacial lake inventory (High Mountain Asia Glacier-lake inventory: Hi-MAG; Chen et al., 2021), which was generated for the entire Himalayan region using data from the 2008–2017 period. We finally chose the 2011 and 2017 proglacial-type lakes (n = 832) and calculated the expansion rates between 2011 and 2017 in the eastern Himalaya region (including the Lunana region).

### 3.4 Emergence velocity and ice floatation of Thorthormi Glacier

We calculated the emergence velocities along Thorthormi Glacier to evaluate the change in glacier dynamics since its detachment from the terminal moraine. We estimated the emergence velocity of a given section ($V_e$, m a$^{-1}$) from the ice fluxes along the upper and lower boundaries of the section as (e.g., Nuimura et al., 2011; Vincent et al., 2016; Brun et al., 2018):

$V_e = \frac{(q_{in} - q_{out})}{\overline{W} \cdot dx},$ (2a)



where $q_{in}$ and $q_{out}$ are the ice fluxes (m$^3$ a$^{-1}$) along the upper and lower boundaries, respectively, and $\bar{W}$ and $dx$ are an averaged glacier width (m) and length (200 m in this study) for analysed $V_e$ section, respectively. The ice flux $q$ ($q_{in}$ or $q_{out}$, m$^3$ a$^{-1}$) is calculated as:


$$q = W \cdot h \cdot V_c, \qquad (3)$$

where $W$, $h$ and $V_c$ are the glacier width (m), ice thickness (m) and depth-averaged ice velocity (m a$^{-1}$), respectively. We then applied a simplified assumption that the glacier width is constant, such that Eq. (2a) can be rewritten as:


$$V_e = \frac{h_{up} \cdot V_{c,up} - h_{low} \cdot V_{c,low}}{dx}, \qquad (2b)$$

where $h_{up/low}$ are the ice thicknesses (m) and $V_{c,up/low}$ are the depth-averaged ice velocities (m a$^{-1}$) along the upper/lower boundaries. We assumed that both the glacier thickness and width were constant in the transverse and longitudinal directions,

respectively, to calculate the emergence velocities along the central flowline. The ice thickness $h$ along the central flowline was calculated from the HP-DEM-derived glacier surface elevation and estimated bedrock elevation in Tsutaki et al. (2019). Tsutaki et al. (2019) estimated the glacier-bed topography using the Farinottie et al. (2009) ice thickness model and tuning a model parameter based on the observed lake depth (ice thickness). We defined the depth-averaged flow velocity ($V_c$) as 80% of the surface velocity (e.g., Sakai et al., 2006; Berthier and Vincent, 2012), and then used the surface velocity component of

the same vector in the central flowline direction. The sections without flow velocities (2520–3020 m from the 2002 terminus) were linearly interpolated using the surface velocities of the surrounding upglacier and downglacier sections. We calculated the emergence velocity for a 200-m section by shifting the section in 50-m increments, and obtained a mean emergence velocity around the current terminus (2400–3500 m from the 2002 terminus). We also calculated $V_e$ in 2011, when Thorthormi Glacier was still a land-terminating glacier, to compare the land- and lake-terminating conditions. We

estimated the ice thickness from the glacier surface elevation of the ASTER-derived DEM acquired on 6 April 2011 and the glacier-bed elevation along the central flowline. The depth-averaged ice velocities were calculated from the surface velocities in Tsutaki et al. (2019) (Sect 3.3).

We evaluated the ice floatation potential of the Thorthormi Glacier terminus based on the ice floatation thickness ($h_f$, m), which was calculated as (Boyce et al., 2007; Watson et al., 2020):


$$h_f = \frac{\rho_w}{\rho_i} d, \qquad (4)$$

where $\rho_w$ is the density of water (1000 kg m$^3$), $\rho_i$ is the density of ice (917 kg m$^3$) (e.g., Boyce et al., 2007; Carrivick et al., 2017) and $d$ is lake depth (m). We then defined an index of potential ice floatation as:

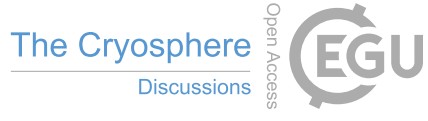


$$P_f = \frac{h_f}{h} \times 100, \qquad\qquad\qquad (5)$$

where $P_f$ is the potential ice floatation (%). The glacier can attain floatation when the glacier ice thickness reaches $h_f$, such that $P_f$ is ≥100%. We extracted the lake surface elevation (4415 m a.s.l.) from the HP-DEM-derived lake perimeter and

estimated the lake depth from the glacier-bed elevation (Tsutaki et al., 2019). We then calculated $P_f$ in 100-m intervals in 2011 and 2018 along the glacier central flowline in the terminus region (up to 3500 m from the 2002 terminus).

## 4 Results

### 4.1 Lake expansion

We traced the lake expansion for the 2000–2018 period (Fig. 2a and 2b). The proglacial lake areas at the termini of

Thorthormi and Lugge glaciers were 3.05 and 1.58 km² in 2018, an increase of 2.01 (193%) and 0.48 km² (44%) from the 2000 lake areas, respectively (Fig. 2c). Both lakes have expanded throughout the study period, and the lake expansion rates (dA/dt) were calculated via a linear regression of the cumulative areas during the 2000–2011 and 2011–2018 periods (Fig. 2c). Lugge Glacial Lake steadily expanding during the 2000–2018 period, with 0.03 and 0.02 km² a⁻¹ observed before and after 2011, respectively. However, there has been accelerated expansion of Thorthormi Glacial Lake since 2011, with 0.07

km² a⁻¹ observed before 2011 and 0.13 km² a⁻¹ observed after 2011. A comparison of these observations with the Hi-MAG data (Chen et al., 2021) indicates that the expansion rates are in the upper 2.5% (Thorthormi) and 10% (Lugge) of the observed proglacial lakes across the eastern Himalayas.

### 4.2 Thinning rates

The dh/dt values of both glaciers were calculated from the 2011 GPS-DEM and 2018 HP-DEM, with the 2002 terminus

position used as the base position for the comparison. We also extracted the calculated dh/dt values from previous studies that had focused on different time periods (Fig. 3 and Table 1). The thinning rate of Lugge Glacier was more than three times greater than that of Thorthormi Glacier for the 2004–2011 period, when Thorthormi Glacier was a land-terminating glacier, and was then comparable (−2.78 m a⁻¹) to that of Lugge Glacier (−2.87 m a⁻¹) for the recent 2011–2018 period, when Thorthormi Glacier had evolved into a lake-terminating glacier (Fig. S1). There was a two-fold increase and 0.61-fold

decrease in the thinning rates of Thorthormi and Lugge glaciers between the 2004–2011 analysis by Tsutaki et al. (2019) and our presented 2011–2018 analysis. The RS-based dh/dt values for the 2000–2016 period (−3.81 to −3.50 m a⁻¹, Brun et al., 2017; Maurer et al., 2019) are similar to the Tsutaki et al. (2019) values (Table 1). The RS-based dh/dt values of Thorthormi and Lugge glaciers are −0.16 and −1.20 m a⁻¹ for the 1975–2000 period, respectively, which suggests that the lower section of Thorthormi Glacier experiences minimal thinning before 2000 (Maurer et al., 2019). The spatial distribution of dh/dt



along Thorthormi Glacier exhibited a decreasing trend in the upglacier direction during the 2011–2018 period, whereas the
dh/dt values during the 2004–2011 period were almost constant across the same region. The thinning rate was >4 m a$^{-1}$
(dh/dt of less than −4 m a$^{-1}$) in the upglacier area (>2500 m from the 2002 terminus) during this later period. The dh/dt
profiles obtained in previous studies do not reveal such a remarkable trend; however, similar dh/dt plots are independent of
the distance from the terminus (Fig. 3a). The results of this study reveal a large spatial variability compared with the RS-

based dh/dt distributions of previous studies (Brun et al., 2017; Maurer et al., 2019), which is likely due to the differences in
the spatial resolution of the data. Tsutaki et al. (2019) reported that Lugge Glacier has a heavily crevassed, bumpy surface.
We consider that the high resolution (0.5 m) of the dh/dt profiles identified the displacements due to these steep surface
features.

## 4.3 Flow velocity

We calculated the surface velocity field between November 2016 and November 2017, and extracted the velocities along the
central flowline (Fig. 4). We also plotted the ITS_LIVE product and Tsutaki et al. (2019) velocity profiles. We found fast
flow velocities (>200 m a$^{-1}$) from the 2017 terminus to the middle part (~4200 m from 2002 terminus) of Thorthormi Glacier.
The ITS_LIVE velocity also exhibited a similar flow-velocity magnitude near the 2017 terminus; however, the ITS_LIVE
velocities decreased more rapidly in the upglacier direction than our calculated results. We are able to confirm the large

displacement of a surface feature (~200 m a$^{-1}$) from the Sentinel-2 satellite images at ~3500–4000 m from the 2002 terminus
(Fig. S2), which suggests that the flow-velocity profile for Thorthormi Glacier that is calculated in this study should be more
reliable than the automatically derived ITS_LIVE flow-velocity profile. A comparison of our velocity profile (2016–2017)
with the 2002–2010 velocity profile (Tsutaki et al., 2019) reveals a substantial 2–4-fold increase at ~2400–4000 m from the
2002 terminus. Tsutaki et al. (2019) projected the flow velocities of Thorthormi Glacier under the assumption of a "lake-

terminating condition" (dashed line in Fig. 4a). Although the magnitude is ~100 m a$^{-1}$ less than that in this study, the
increasing flow velocities toward the calving front are similar to the trend in the recent velocity profile.

  Conversely, the calculated surface velocities of Lugge Glacier are ~50 m a$^{-1}$ up to ~2000 m from the 2002 terminus in this
study. There also appears to be a gradual decrease up to ~2700 m from the 2002 terminus (Fig. 4b). The ITS_LIVE velocity
profile possesses <5 m a$^{-1}$ flow velocities for the entire glacier, which is probably due to the coarser resolution (240 m) of

the velocity field compared with that in this study (10 m) and Tsutaki et al. (2019; 15 m). Although the terminus position of
Lugge Glacier has retreated almost 1 km since 2002, the mean velocity profile appears to have remained persistent between
the 2000–2010 (Tsutaki et al., 2019) and 2016–2017 observation periods.

## 4.4 Ice emergence velocity and floatation potential of Thorthormi Glacier

We calculated the emergence velocity of the Thorthormi Glacier terminus as −0.6 m a$^{-1}$ (2400–3500 m from the 2002

terminus), although there are large variations depending on the computational area (σ = ±10.6 m a$^{-1}$). $V_e$ was 4.6 ± 3.4 m a$^{-1}$
for the land-terminating condition, which suggests that the mean $V_e$ has decreased and become negative after transitioning to



a lake-terminating glacier. Tsutaki et al. (2019) also estimated $V_e$ via numerical modelling of the lake- and land-terminating conditions, yielding $2.2 \pm 1.9$ and $6.3 \pm 2.2$ m a$^{-1}$, respectively.

We also estimated the potential ice-floatation index in the terminus area of Thorthormi Glacier (up to 3500 m from the
2002 terminus). The mean $P_f$ values for 2011 (land-terminating) and 2018 (lake-terminating) are 86% and 97%, respectively, with this increase attributed to the surface lowering of the terminus area during the 2011–2018 period. As a result of surface lowering, some parts of the ice in the terminus area reached ice-floatation thickness ($P_f > 100\%$) by 2018.

## 5 Discussion

Thorthormi and Lugge glaciers possess contrasting dh/dt trends and flow velocities, even though they are adjacent to each
other. The dh/dt trend, and flow velocity magnitude and spatial distribution of Lugge Glacier are approximately constant between the 2004–2011 and 2011–2018, and 2002–2010 and 2016–2017 periods, respectively (Figs. 3b and 4b). Conversely, remarkable increases in the thinning rate and flow velocity are observed across Thorthormi Glacier over the same study periods (Figs. 3a, 4a and 5). Such a drastic velocity increase within a decade has not been reported in the Himalayas, although the multi-decadal acceleration of glacier thinning and deceleration of glacier flow have been reported (Dehecq et al.,
2019; Maurer et al., 2019).

Tsutaki et al. (2019) performed finite-element simulations of present (land-terminating) and future (lake-terminating) Thorthormi Glacier dynamics. Their simulations reproduced the flow velocities for the land-terminating condition with a small root-mean-square error ($<10$ m a$^{-1}$) using satellite-based flow velocities. Their future prediction for a lake-terminating condition, which suggested an increase in flow velocity, is inconsistent with our 2017 velocity analysis (Fig. 4a). However,
Tsutaki et al. (2019) highlighted that changes to the sliding coefficient and ice thickness parameters could alter the flow velocity significantly, as their sensitivity tests demonstrated that the simulated flow velocity increased (decreased) by 33% (51%) if the sliding coefficient and ice thickness changed by +30% (−30%) for the land-terminating condition of Thorthormi Glacier. Therefore, the difference between the observed and simulated velocities is likely due to the uncertainties in the sliding coefficient, ice thickness and state of the terminus position. Despite this underestimation, Tsutaki et al. (2019) have
reasonably demonstrated the change in sliding conditions associated with the transition from land- to lake-terminating conditions.

Proglacial lakes had formed on both sides of the Thorthormi Glacier terminus before 2011. The ice thickness was near flotation ($P_f > 85\%$), such that the flow velocities could accelerate near the terminus. However, the flow velocities decreased toward the terminus, producing a longitudinal compression field and subsequent surface lowering that might have been less
than that of lake-terminating Lugge Glacier (Tsutaki et al., 2019). The longitudinal stress field regime changed from compressional to extensional after the terminus detached from the terminal moraine and transitioned to a lake-terminating condition in 2011, and its flow increased owing to efficient basal sliding. Furthermore, the lakes that formed on both sides of the glacier terminus may also have reduced the lateral resistive stresses that prevented glacier flow (e.g., Adhikari et al.,



2012). These factors might have led to the observed dramatic increase in flow velocities (Fig. 4a). Such an increase in flow
velocities due to proglacial lake formation has been observed in other regions (e.g., Boyce et al., 2007; Tsutaki et al., 2011;
Sakakibara and Sugiyama, 2014); however, this is the first observation of such a phenomenon in the Himalayan region.

The rapid increase in flow velocities may have enhanced the ice flux towards the glacier terminus due to the longitudinal
strain. Positive emergence velocities are distributed up to 2400–3500 m from the 2002 terminus for the 2011 land-
terminating condition (Sect 4.4). However, $V_e$ decreased in 2017 and became negative due to the increase in flow velocities
toward the terminus. The unchanged thinning rate and velocity regime of Lugge Glacier (Figs 2b and 3b, and Table 1)
suggest that any recent climatic changes in the Lunana region could not have yielded a significant increase in surface
ablation. Therefore, the increased thinning rate of Thorthormi Glacier can be largely attributed to the decrease in $V_e$.
However, the decrease in $V_e$ ($-5$ m a$^{-1}$) seems to be too large to account for the increased thinning rate from 2004–2011 to
2011–2018 ($-1.38$ m a$^{-1}$; Table 1). This decrease may be due to the uncertainty in the estimated ice thickness; regardless,
this decreasing $V_e$ trend can be clearly confirmed. We simply assumed a constant glacier width to calculate $V_e$ along the
central flowline. However, this glacier terrain tends to widen in the downglacier direction, yielding an extensional velocity
regime. The lateral proglacial lakes on both sides of the terminus before it transitioned to a lake-terminating condition may
have further contributed to a more negative $V_e$ than that estimated along the central flowline. Despite these favourable
conditions to enhance dynamic thinning, the surface lowering of Thorthormi Glacier has likely been suppressed by the
compressive flow regime of the land-terminating condition. The transition to a lake-terminating condition should have
caused a two-fold increase in the thinning rate during such a short period (Fig. 2a and Table 1). These above-mentioned
mechanisms might cause a positive feedback between glacier thinning and the increase in flow velocity by enhancing each
other. Therefore, increased glacier thinning and surface velocity speed-up will continue along Thorthormi Glacier in the
future. The dynamic thinning of lake-terminating glaciers has been discussed in other HMA regions (e.g., Nuimura et al.,
2012; King et al., 2018; Liu et al., 2020). However, our study is the first reported observation of the dynamic changes during
the transition from land- to lake-terminating conditions, which have led to the enhanced thinning of a Himalayan glacier.

## 6 Conclusion

We presented the surface elevation and velocity changes, and proglacial lake expansion of lake-terminating Thorthormi and
Lugge glaciers in the Lunana region, Bhutanese Himalaya. We analysed satellite/photogrammetry data and compared our
results with those in previous studies to reveal the recent glacier and proglacial lake changes of Thorthormi Glacier, which
are associated with the transition from land- to lake-terminating conditions. Whilst the lake expansion and surface lowering
rates of Luge Glacier have been approximately constant since 2000, those of Thorthormi Glacier have exhibited a continue
increase after the terminus reached floatation and detached from the terminal moraine. There has been a two-fold increase in
the thinning rate of Thorthormi Glacier since this transition to lake-terminating conditions in 2011. The flow-velocity field of
Thorthormi Glacier has also sped up considerably ($>150$ m a$^{-1}$), whereas that of Lugge Glacier has remained unchanged. We



estimate that the rapid thinning and increased flow-velocity field of Thorthormi Glacier were due to this transition to lake-terminating conditions. This study provides the first evidence of the dynamic glacier changes associated with proglacial lake formation in the Himalayan region, and will contribute to advancing our understanding of the dynamics of lake-terminating glaciers, as well as their potential evolution in the future.


*Data availability.* The Landsat 7 ETM+, Landsat 8 OLI, and Sentinel-2 satellite data are distributed by the United States Geological Survey (https://earthexplorer.usgs.gov/, last access: 19 October 2021). ASTER-DEM data are distributed by the National Institute of Advanced Industrial Science and Technology (https://gbank.gsj.jp/madas/map/index.html, last access: 19 October 2021).


*Competing interests.* The authors declare that they have no conflict of interest.

*Author contributions.* KF designed the study. HI conducted the photogrammetry survey, with the help of KK. YS processed the photogrammetry data and analysed the satellite data. YS, KF and AS wrote the manuscript. All of the authors contributed
to the discussion.

*Acknowledgment.* We thank J. Chopel and S. Ohmi for supporting the aerial photogrammetry survey. We are indebted to S. Tsutaki and T. Nuimura for providing their data and supporting our data analysis. We thank E. Berthier for providing the Pléiades satellite data.

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

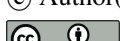



**Table 1.** Comparison of the measured surface elevation change rates of Thorthormi and Lugge glaciers from various studies.

| Rate of surface elevation change (m a$^{-1}$) | | | |
|---|---|---|---|
| **Period** | **Thorthormi Glacier** | **Lugge Glacier** | **Reference** |
| 1975–2000 | −0.16 | −1.20 | Maurer et al. (2019) |
| 2000–2016 | −1.30 | −3.50 | Maurer et al. (2019) |
| 2000–2016 | −1.29 | −3.81 | Brun et al. (2017) |
| 2004–2011 (DGPS) | −1.40 ± 0.27 | −4.67 ± 0.27 | Tsutaki et al. (2019) |
| 2004–2011 (ASTER) | −1.61 ± 2.75 | −2.24 ± 2.75 | Tsutaki et al. (2019) |
| 2011–2018 | −2.78 ± 0.62 | −2.87 ± 0.62 | This study |



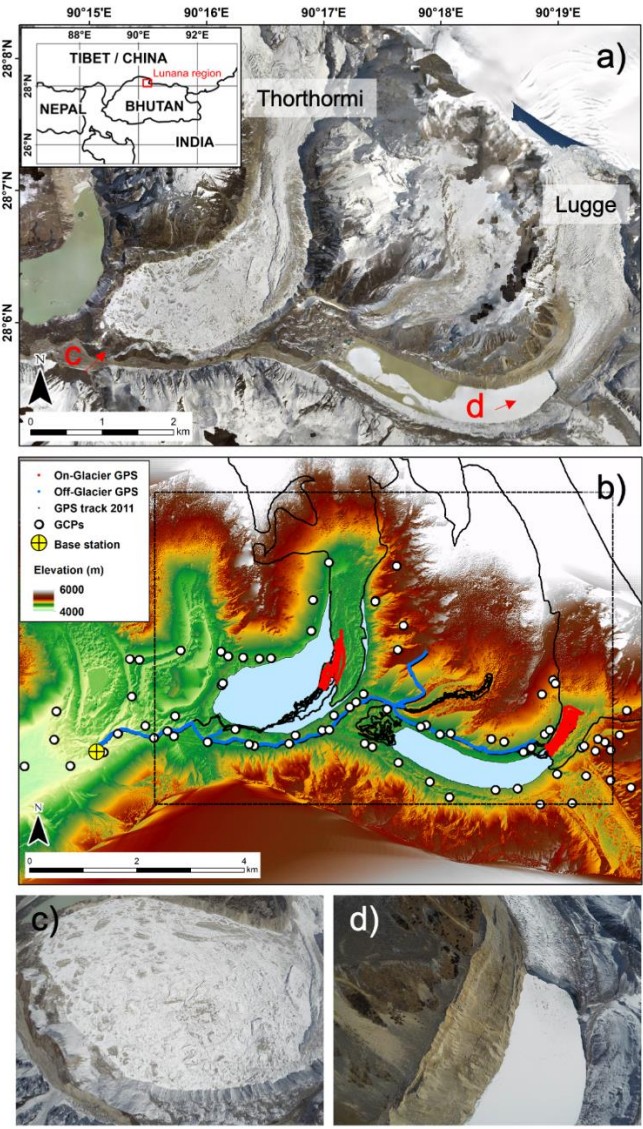

**Figure 1.** Details of the study site. (a) Location of the Lunana region (inset) and helicopter photogrammetry (HP) orthoimage of Thorthormi and Lugge glaciers (acquired on March 24, 2018). (b) Surface elevation map generated from the HP-DEM using ground control points (GCPs) for terrain data processing (open circles) and 2011 GPS tracks (dots). (c) and (d) Aerial photographs of Thorthormi and Lugge glaciers. Red arrows in panel (a) indicate the directions from which the aerial photographs were taken. The dashed box in (b) shows the domain of (a). Red and blue GPS tracks in (b) are used for the elevation change analysis and DEM accuracy assessment (Sect. 3.2), respectively; the black tracks are not used for the elevation change analysis.



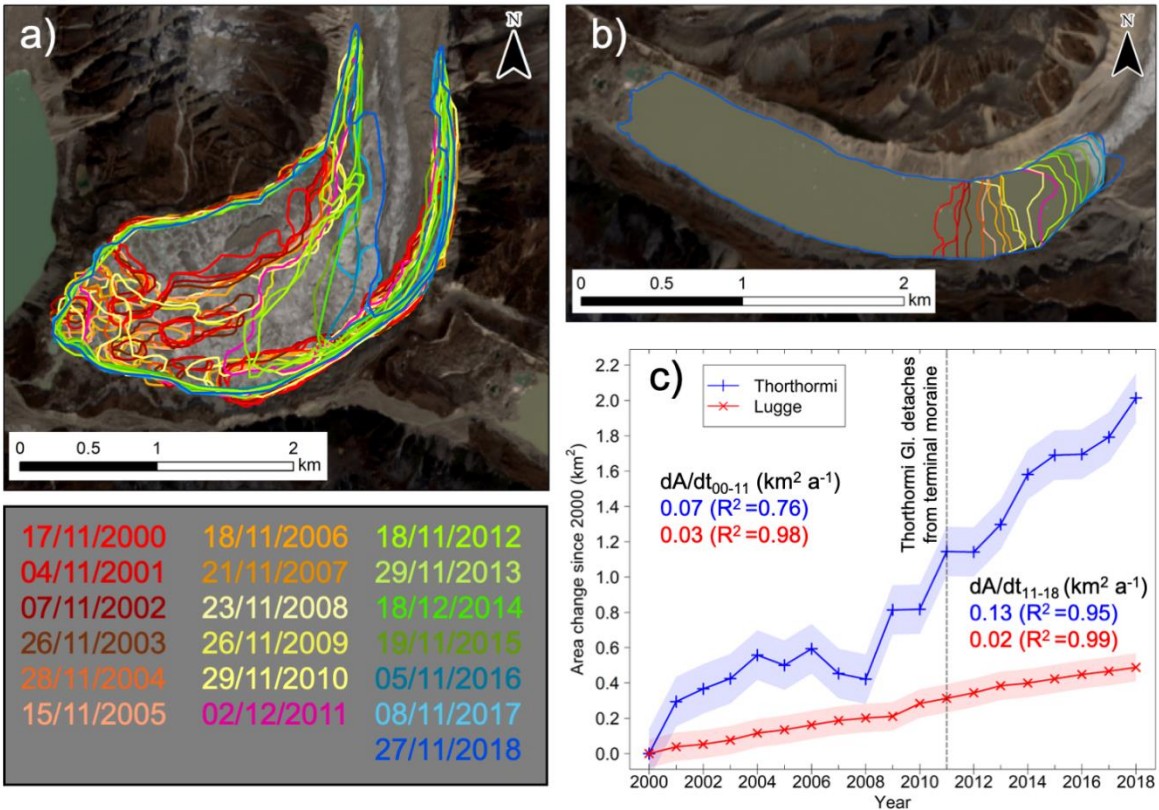

**Figure 2.** Temporal variations in the spatial extents of (a) Thorthormi and (b) Lugge proglacial lakes. (c) Cumulative changes in lake area for Thorthormi and Lugge glaciers relative to 2000. Background images of (a) and (b) are Sentinel-2 satellite images that were acquired on 11 November 2017. The dA/dt values in (c) are the 2000–2011 (upper left) and 2011–2018 (lower right) lake expansion rates.





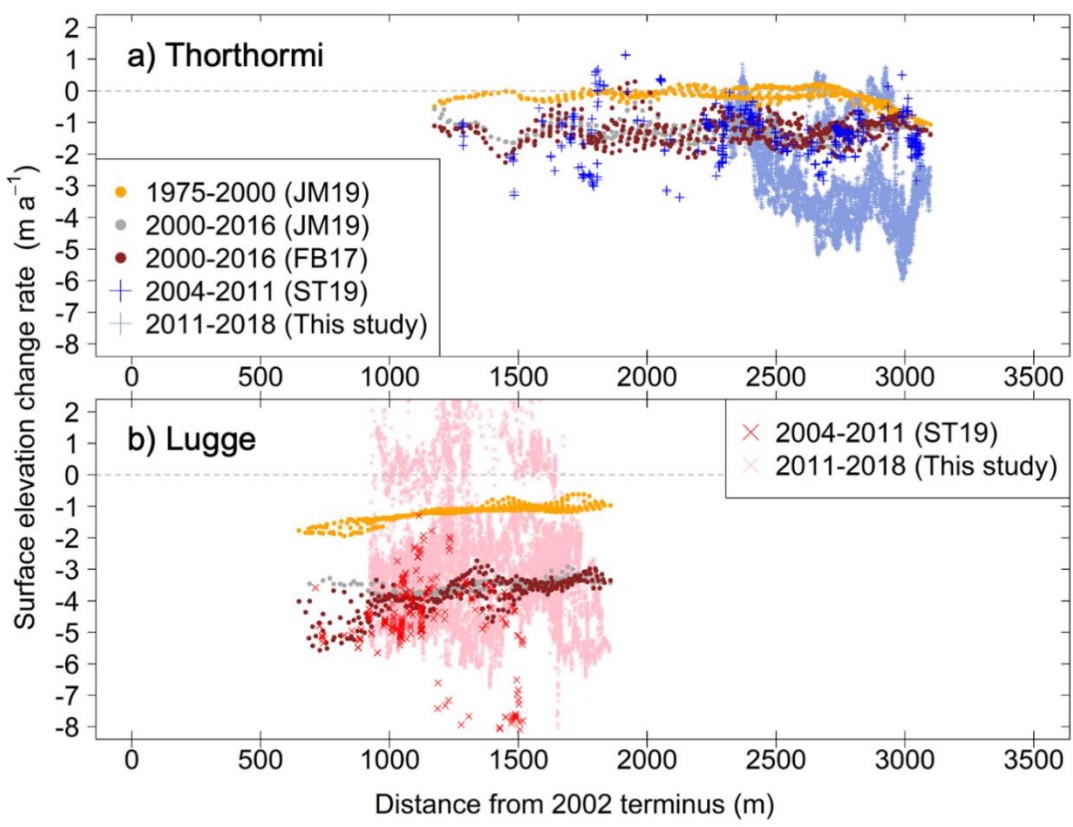

**Figure 3.** Surface elevation change rates (dh/dt) along (a) Thorthormi and (b) Lugge glaciers (based on the distance from the 2002 glacier terminus). Each panel shows the elevation change rates for 1975–2000 and 2000–2016 (Maurer et al., 2019; JM19), 2000–2016 (Brun et al., 2017; FB17), 2004–2011 (Tsutaki et al., 2019; ST19) and 2011–2018 (this study).

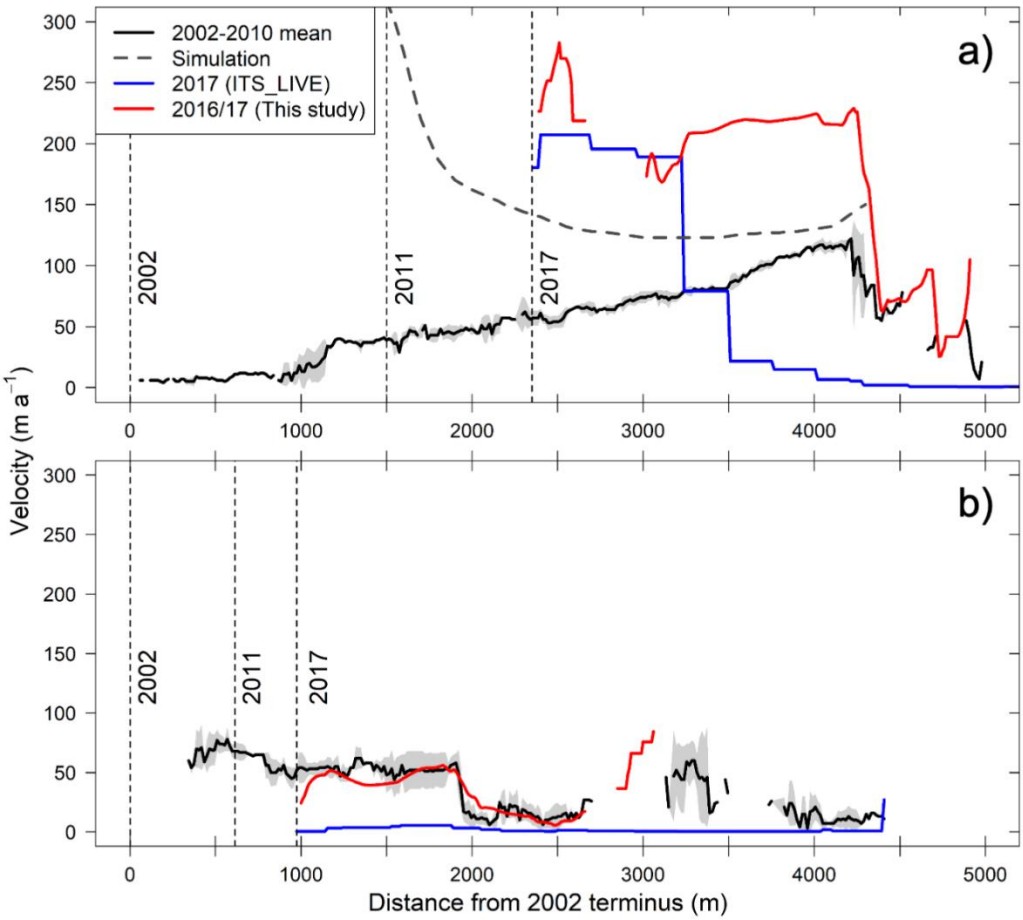

**Figure 4.** Central flowline velocities of (a) Thorthormi and (b) Lugge glaciers. Dashed vertical lines indicate the glacier terminus positions in 2002, 2011 and 2017. Blue and red lines represent the flow velocities from the ITS_LIVE surface velocity field and this study, respectively. Black lines and grey shaded regions represent the mean and standard deviation of the flow velocities for the 2002–2010 period, respectively (Tsutaki et al., 2019). The thick dashed line in (a) denotes simulated surface velocities with a lake-terminating assumption (Tsutaki et al., 2019).