# Peer review of "Land- to lake-terminating transition triggers dynamic thinning of a Bhutanese glacier"

_The Cryosphere, 2021_

## Author Comment (AC1)

**Response to referee #1**

We thank the referee for the comments on our manuscript. We answer all comments in blue font.

**General comments:**

The article provides a very good understanding of the topic that gives the reader an overview of glacier retreat and subsequent proglacial lake formation across the Himalayas region. The research was well conducted, and information was presented accordingly.

Thank you for your comments.

The research observed the dynamic changes in two glaciers in the Bhutanese Himalaya, which should reflect on the article's title.

Thank you for your useful suggestion. However, the main aim of this study is to trace the fluctuations of the Thorthormi glacier (land- to lake-terminating transition). Lugge Glacier was shown as a "reference glacier" to contrast Thorthormi Glacier. Therefore, we would like to keep this title.

The methodology and data analysis has been well explained in each section, but the overall methodology of how different sets of data (with different source and resolution) are compared and analyzed would help readers to get a better understanding.

We received similar comments from referee #2 (regarding dh/dt), so we will change Fig.3, which shows elevation changes based on resampled DEMs (all 30 m resolutions).

In terms of surface flow velocities, we also used different resolutions datasets. We had described the effect of different resolution in the result section (lines 243-245) as "The ITS_LIVE velocity profile possesses <5 m a$^{-1}$ flow velocities for the entire glacier, which is probably due to the coarser resolution (240 m) of the velocity field compared with that in this study (10 m) and Tsutaki et al. (2019; 15 m))."

[Figure]

**Figure 3** revised.

**Technical corrections:**

"ITS_LIVE" has not been explained through the article.

We will add an explanation about "ITS_LIVE" in lines 139-142 as "We extracted the surface velocity from regional velocity product derived from Inter-Mission Time Series of Land Ice Velocity and Elevation (ITS_LIVE) project (Gardner et al., 2019), which covered the entire HMA region. The ITS_LIVE velocity product is generated from Landsat 4, 5, 7, and 8 with the auto-RIFT feature tacking processing chain (Gardner et al., 2018) and possesses a 240-m spatial resolution.".

We will add reference "Gardner, A. S., Moholdt, G., Scambos, T., Fahnstock, M., Ligtenberg, S., van den Broeke, M., and Nilsson, J.: Increased West Antarctic and unchanged East Antarctic ice discharge over the last 7 years, cryosphere, 12, 521–547, https://doi.org/10.5194/tc-12-521-2018, 2018."

"GAMDAM" has not been explained through the article.

We will add an explanation of GAMDAM glacier inventory and change lines 62-65 as "To determine the glacier outlines, we employed "Glacier Area Mapping for Discharge from the Asian Mountains" inventory (GAMDAM glacier inventory; Nuimura et al., 2015; Sakai et al., 2019) covering throughout high-mountain Asia."

---

## Author Response (AR1)

**Author's response**

**Response to referee #1**
We thank the referee for the comments on our manuscript. We answer all comments in blue font.

**General comments:**
The article provides a very good understanding of the topic that gives the reader an overview of glacier retreat and subsequent proglacial lake formation across the Himalayas region. The research was well conducted, and information was presented accordingly.
Thank you for your comments.

The research observed the dynamic changes in two glaciers in the Bhutanese Himalaya, which should reflect on the article's title.
Thank you for your useful suggestion. However, the main aim of this study is to trace the fluctuations of the Thorthormi glacier (land- to lake-terminating transition). Lugge Glacier was shown as a "reference glacier" to contrast Thorthormi Glacier. Therefore, we would like to keep this title.

The methodology and data analysis has been well explained in each section, but the overall methodology of how different sets of data (with different source and resolution) are compared and analyzed would help readers to get a better understanding.
We received similar comments from referee #2 (regarding dh/dt), so we will change Fig.3, which shows elevation changes based on resampled DEMs (all 30 m resolutions).
In terms of surface flow velocities, we also used different resolutions datasets. We had described the effect of different resolutions in the result section (lines 243-245) as "The ITS_LIVE velocity profile possesses <5 m a$^{-1}$ flow velocities for the entire glacier, which is probably due to the coarser resolution (240 m) of the velocity field compared with that in this study (10 m) and Tsutaki et al. (2019; 15 m)."

[Figure]

**Figure 3** revised.

**Technical corrections:**

"ITS_LIVE" has not been explained through the article.

We will add an explanation about "ITS_LIVE" in lines 138-141 of the revised manuscript "We extracted the surface velocity from the regional velocity product derived from the ITS_LIVE (Inter-Mission Time Series of Land Ice Velocity and Elevation) project (Gardner et al., 2019), which covered the entire HMA region. The ITS_LIVE velocity product is generated from the Landsat series with the auto-RIFT feature tracking processing chain yielding a 240-m spatial resolution (Gardner et al., 2018).".

We will add reference "Gardner, A. S., Moholdt, G., Scambos, T., Fahnstock, M., Ligtenberg, S., van den Broeke, M., and Nilsson, J.: Increased West Antarctic and unchanged East Antarctic ice discharge over the last 7 years, cryosphere, 12, 521–547, https://doi.org/10.5194/tc-12-521-2018, 2018."

"GAMDAM" has not been explained through the article.

We will add an explanation of the GAMDAM glacier inventory and change lines 64-65 in the revised manuscript as "Thorthormi Glacier covers 11.6 km$^2$ based on the GAMDAM (Glacier Area Mapping for Discharge from the Asian Mountains) glacier inventory (Nuimura et al., 2015; Sakai, 2019) and the 2017 terminus position."

We thank the referee for the comments on our manuscript. We answer all comments in blue font.

**General comments:**

Sato and colleagues present a short study on the effects of a Himalayan glacier retreat with a transition of termini type from a land-terminating to lake-terminating. They argue that the recently observed significant acceleration of ice flow velocities of Thorthormi glacier can be explained by a switch from a compressional to an extensional flow regime in the ablation area caused by a detachment from the terminal moraine. These findings are supported by analysis of ice flow velocity and ice elevation change and are built upon the results of a previously published study by Tsutaki et al. (2019). Thank you for your comments.

Unfortunately, at the moment the novelty of this submission is fairly limited as the major points have already been addressed by Tsutaki et al. (2019) and by the recent regional study of Pronk et al. (2021). As far as I'm concerned, the purpose of the paper is to show the observational data that confirm the Tsutaki et al. (2019) hypothesis of a plausible ice flow acceleration of Thorthormi glacier following a detachment from the terminal moraine. As the authors have noticed, similar acceleration has already been observed in other regions and, according to current understanding, is something likely to happen.

We first emphasize that this comment doesn't evaluate our study appropriately. This study aims NOT to compare land- and lake-terminating glaciers BUT to trace the fluctuation of a single glacier associated with its land- to lake-terminating transition. Therefore, the novelty of this study would not be discredited by Pronk et al. (2021) and related previous studies, which compared different terminus types. Although a few previous studies traced such the glacier fluctuations associated with proglacial lake formation and terminus detachment in Alaska and Europe Alps (Boyce et al., 2007; Tsutaki et al. 2011), the number of cases and the time periods are very limited compared with those of regional studies dealing with the different terminus types. Furthermore, this is surely the first study that witnessed the terminus transition and associated changes in glacier behavior in the Himalayan region. We had mentioned this in lines 304-306 as "The dynamic thinning of lake-terminating glaciers has been discussed in other HMA regions (e.g., Nuimura et al., 2012; King et al., 2018; Liu et al., 2020). However, our study is the first reported observation of the dynamic changes during the transition from land- to lake-terminating conditions, which have led to the enhanced thinning of a Himalayan glacier.".

Unfortunately, the authors did not make a sufficient effort to show the new data convincingly. Figures with maps of ice elevation change and ice velocity field or a profile of the emergence velocity would have been useful and could give a quantitative grasp on what is happening during the retreat. Secondly, the observations of the glacier speed up are limited to only one season (2016/2017 - five years ago), it would have been much more interesting to see a time series of the annual velocities covering the full period from 2011 to the present.

Moreover, the calculation of the emergence velocities is flawed as the authors incorrectly assume a constant no-slip condition (mean vertical velocity equal to 80% of the surface velocity) along the entire ablation area of the glacier.

To reply to these comments, we will add the following analyses and discussion.

- **Annual surface velocity changes**

The limited flow velocity is due to the satellite we used (Sentinel-2 available since 2016). To trace the temporal change in flow velocity, we adopted the ITS_LIVE annual velocity data (2010–2018). We extracted velocity data every 250 m along the glacier center flowlines. ITS_LIVE velocities do not seem to represent the flow velocity of Lugge Glacier (revised Fig. 4b), while those of Thorthormi Glacier seem well represent the annual change (revised Fig. 4a).

[Figure]

**Figure 4** revised.

Acknowledging this comment, we realized some important points that were not in the original manuscript. We thought that the Thorthormi terminus detached in 2011, and velocity soon increased after that. However, the ITS_LIVE velocities

increased drastically after 2017 while those before 2017 show a similar profile to those estimated by ASTER data (Tsutaki et al., 2019). This suggests that the Thorthormi terminus might have been land-terminating condition underwater though the proglacial lake had formed since 2011. We therefore consider that the subaqueous terminus detached from the terminal moraine/ice during 2016–2017, and then the drastic speed-up of velocities occurred. We sincerely thank reviewer #2 for this valuable comment indeed despite the overall negative evaluation. From this finding, we re-evaluated changes in emergence velocity and its impact on the surface lowering of Thorthormi Glacier as follows.

- **Emergence velocity calculation**

We agree with the comment that our assumption was not suitable for estimating the emergence velocity. Because Tsutaki et al. (2019) simulated that the basal velocity reaches 90% of the surface velocity of Thorthormi, we re-calculated emergence velocity with the two assumptions that the vertical mean velocity equals to 90% (assumption by Tsutaki et al., 2019) and 100% (assumption of a floating condition after terminus detachment) of the surface velocity. We mentioned it in lines 178-281 in the revised manuscript as "Tsutaki et al. (2019) simulated that the basal velocity reaches 90% of the surface velocity of Thorthormi Glacier. We therefore calculated the emergence velocity using two assumptions regarding the surface velocity: the depth-averaged velocity is 90% of the surface velocity based on Tsutaki et al. (2019), and 100% assumed for a floating condition after terminus detachment."

Although the longitudinal profiles of emergence velocity are largely variable probably due to uncertain ice thickness and other simplified assumptions (below Supplementary Figure for emergence velocity), the mean values for the target domain are obviously contrasted from positive to slightly negative (Table 2). We will add a figure of the emergence velocity profile and the mean emergence velocities. In the following reply, we further quantified the impact of drastic change in the emergence velocity on the surface lowering.

**Table 2.** Comparison of the emergence velocity of Thorthormi Glacier in 2011 and 2017. The mean values are calculated for the 2400–3500 m section from the 2002 terminus (Fig. S4). Two basal sliding conditions are assumed, whereby depth-averaged velocity equals either 90% or 100% of the surface velocity.

| | 2011 | | 2017 | |
|---|---|---|---|---|
| Depth averaged-velocity | 90% | 100% | 90% | 100% |
| Emergence velocity (m a$^{-1}$) | $5.20 \pm 3.78$ | $5.78 \pm 4.20$ | $-0.69 \pm 11.65$ | $-0.77 \pm 12.94$ |

[Figure]

**Supplementary Figure.** Emergence velocities along the center flowline of Thorthormi Glacier. The mean emergence velocity ($V_e$) is calculated for the 2400-3500 m section where the dh/dt data is available (Table 2).

● **Additional discussion in emergence velocity**

In Thorthormi Glacier, the emergence velocity decreased by more than 4 m a$^{-1}$ from the 2011 land-terminating condition to the 2018 lake-terminating condition. It seems too large to account for the increased thinning rates (1.38 m a$^{-1}$) from 2004–2011 (–1.40 m a$^{-1}$) to 2011–2018 (−2.78 m a$^{-1}$) (Table 1). We quantified the change in emergence velocity by hypothesizing that it occurred in the last two years (2017 & 2018) during which the surface velocity accelerated. Based on this hypothesis, we calculated the weighted average of the emergence velocity for the period 2011–2018 ($V_{e,2011-2018}$, m a$^{-1}$) as:

$$V_{e,2011-2018} = \frac{\Delta t_{\text{land}} V_{e,\text{land}} + \Delta t_{\text{lake}} V_{e,\text{lake}}}{\Delta t_{\text{land}} + \Delta t_{\text{lake}}}$$

Here, under the assumption that depth-averaged ice velocity is equal to 90% of surface velocity (Table 2), $V_e$ and $\Delta t$ are the emergence velocity and years of the land- and lake-terminating conditions, respectively. With the emergence velocities ($V_{e,\text{land}} = 5.20$ m a$^{-1}$ and $V_{e,\text{lake}} = -0.69$ m a$^{-1}$), and periods ($\Delta t_{\text{land}} = 5$ and $\Delta t_{\text{lake}} = 2$), we then obtained the time-weighted mean emergence velocity for the period 2011–2018 as 3.52 m a$^{-1}$. It means that the emergence velocity reduced by –1.68 m a$^{-1}$ before and after 2011, and it is well consistent with the change in dh/dt from 2004–2011 to 2011–2018 (−1.38 m a$^{-1}$). Largely variable profiles of the emergence velocity suggest that the estimates are highly uncertain. However, our first order evaluation can explain the cause of the drastic change in the thinning rate of Thorthormi Glacier. We add the evaluation above mentioned to lines 309-321 of the revised manuscript.

We thank refree#2 again for the valuable comments, by which we came up with this idea.

- **Figures of elevation change and surface velocity**
We add the surface elevation change rate in Figure 1b.

[Figure]

**Figure 1b** revised.

We also show the surface velocity field. Although the surface velocity along the glacier center flowline is well calculated, many errors are shown on both sides of the Thorthormi Glacier due to the calving of icebergs. Therefore, we add this figure (velocity map) to the supplementary material as it doesn't help readers understand well.

[Figure]

**Supplementary Figure.** Surface flow velocity field between 16 November 2016 and 11 November 2017. The areas enclosed by the white polygons are stable-ground used to evaluate the error in surface flow velocity.

Lastly, the authors disregard frontal ablation flux in their analysis whereas Pronk et al. (2021) have shown that frontal ablation is controlling the strain rate and whether the terminus is in compressional or an extensional regime. In my opinion, at this stage, the manuscript lacks sufficient quality and novelty that would ensure publication in The Cryosphere and needs a very substantial revision to be accepted.

Thank you for your useful comments. The numerical experiment on the relationship between terminus situation and ice velocity in Pronk et al. (2021) is interesting and informative. However, the effect simulated by Pronk et al. (2021) was not validated by any observational data, and they set a hypothetical terminus (calving front). Although Minowa et al. (2021) quantified such contributions of frontal ablation and subaqueous melting based on their intensive observations in Patagonia, it is not possible to do so with the data available for the glaciers discussed in this study. Whatever frontal ablation or subaqueous melting, the accelerated velocity and estimated emergence velocity suggest that the dynamic thinning should surely account for the accelerated thinning rate of Thorthormi Glacier.

We mentioned the importance of terminus condition with refereeing Pronk et al. (2021) in discussion (Sect. 5) below (lines 335-339 in the revised manuscript).

"This study employed modelled ice thickness (lake depth) that was tuned using point measurement data (Tsutaki et al., 2019) to estimate the dynamics of Thorthormi Glacier. Previous studies have suggested that the surface flow velocity of lake-terminating glaciers is sensitive to the terminus ice thickness and lake water depth (Benn et al., 2007; Pronk et al., 2021). Therefore, constraints on the lake bathymetry may allow us to better understand past and current terminus conditions and quantify the dynamic thinning process. "

We also add Pronk et al. (2021) in the introduction (Sect. 1) as a previous study focusing on a regional scale analysis. We will mention it in line 40 (after the introduction for King et al., 2019) as " Pronk et al. (2021) analysed the surface flow velocities of more than 300 glaciers in the Himalayan region, and determined that the flow velocities of lake-terminating glaciers were twice as high as those of land-terminating glaciers on average."

**Specific comments:**

**L41-42:** This is true not only for land- vs. lake-terminating glaciers, it is not uncommon to see a different response to climate forcing even when the glaciers share the same terminus type.

We agree with your comment. We changed it as "The existence of a proglacial lake might be a factor enhancing the glacier flow velocity, retreat and thinning of HMA glaciers. The response of lake- and land-terminating glaciers can fluctuate with different patterns, even if they are located near each other and/or exist under similar climatic conditions (Liu et al., 2020)."

**L62:** "with the 2017 terminus position and its elevation range spanning 4,400–6,900 m above sea level (a.s.l.)" please rewrite

We will change lines 64-66 of the revised manuscript as "Thorthormi Glacier covers 11.6 km$^2$ based on the GAMDAM (Glacier Area Mapping for Discharge from the Asian Mountains) glacier inventory (Nuimura et al., 2015; Sakai, 2019) and the 2017 terminus position. Its elevation range spans 4,400–6,900 m above sea level (a.s.l.)." to avoid confusion.

**L65:** Which year does this area refer to?

The 2017 terminus position was referred. We will change line 65 on the manuscript as " Lugge Glacier is located to the east of Thorthormi Glacier, and covers an area of 10.0 km$^2$ based on its 2017 terminus position, with its elevation range spanning 4,500–6,900 m a.s.l. ".

**L66:** Please be consistent with lake names and introduce them on the Study Area map: here you refer to Lugge Glacial Lake whereas before, (L65) you introduced Lugge Tscho

We fixed the proglacial lake names as "Lugge Glacier lake/Thorthormi Glacier lake". We will change line 67 as "Lugge glacier lake has expanded since the 1960s (Komori, 2008),…" And we will add proglacial lake names in Figure 1b. Please see the revised Figure 1b in reply to the general comment.

**L116:** If the dh/dt values are at the same positions, how do you explain such a large variance in your dataset compared to previous estimates (Fig. 3)? Did you aggregate your DEMs to 30m resolution as well? Otherwise, both datasets cannot be directly compared.

The high-resolution HP-DEM processed in this study can estimate precise elevation changes compared to the previous works (Brun et al., 2017; Maurer et al., 2019; 30 m-resolution). Therefore, we calculated and plotted surface elevation change without resampling GPS-DEM and HP-DEM. We will change Figure 3 to show elevation changes based on resampled DEMs.

The 2011–2018 elevation changes include positive values. Tsutaki et al. (2019) reported that Lugge Glacier terminus has a heterogeneous surface, which we can confirm from photogrammetry data (Fid. 1c). Therefore, we consider that a displacement of such topography caused positive elevation changes.

We had already mentioned it in lines 224-228 as "The results of this study reveal a large spatial variability compared with the RS-based dh/dt distributions of previous studies (Brun et al., 2017; Maurer et al., 2019), which is likely due to the differences in the spatial resolution of the data. Tsutaki et al. (2019) reported that Lugge Glacier has a heavily crevassed, bumpy surface. We consider that the high resolution (0.5 m) of the dh/dt profiles identified the displacements due to these steep surface features."

[Figure]

**Figure 2** revised**.**

Additionally, the Leibnitz notation (dh/dt) is normally used for instantaneous values (differential) implying that both dh and dt are infinitesimals.

Thank you for your comment. This style (dh/dt) is commonly used in other publications in glacier study (e.g., Brun et al., 2017; Nature Geoscience, King et al., 2017; The Cryosphere). Hence we will use this style.

**L118-122:** Please provide more details on data processing (template size, templatematch algorithm)

We chose 20 pixels (200 m) as a template size and used the standard normalized cross-correlation (NCC) algorithm (Heid and Kääb, 2012). We will explain in the method section (Sect. 3.3).

Lines 120-122 in the revised manuscript;

"The normalized cross-correlation algorithm (NCC; Heid and Kääb, 2012) in the feature tracking toolbox (Templatematch) identifies the displacement patterns of the glacier surface features and computes their magnitude from a pair of images."

Lines 123-125 in the revised manuscript;

" After visual trial and error, we chose a $20 \times 20$ template size ($200 \times 200$ m) and $75 \times 75$ search window size ($750 \times 750$ m) to compute the surface feature displacements and calculate the annual flow velocity.".

We will also add Heid and Kääb, (2012) to the reference list.

"Heid, T. and Kääb, A.: Evaluation of existing image matching methods for deriving glacier surface displacements globally from optical satellite imagery, Remote Sens. Environ., 118, 339–355,

https://doi.org/10.1016/j.rse.2011.11.024, 2012."

**L135-136:** Did you look on ITS_LIVE scene-pair velocities as well?

We had checked scene-pair velocities (120 m resolution). It appears like this figure if we plot scene-paire velocity (hashed black lines) with other annual ITS_LIVE data. The scene-pair velocity profile seems to be similar to other years.

[Figure]

**L149:**flotation

We will change all "floatation" to "flotation" and fix it. Thank you.

**L155:** Why is this Eq. 2a and not 2?

Eq.2a and Eq.2b originate from the same equation so that we denoted as such. We will change (2a) to (2) and (2b) to (4) to avoid confusion.

**L169-170:** How reliable is the assumption of a fixed glacier width and thickness? Please remember that the glacier bed geometry is inferred from inverse modeling (Farinotti et al., 2009; Tsutaki et al., 2019) and has not been measured.

We have set this assumption for two reasons. (1) One of our aims is to compare with the results of Tsutaki et al. (2019). They used the same assumption (2-dimensional model) and calculated emergence velocity on center flow lines. Therefore, changing the assumptions does not make an accurate comparison with previous work. (2) Thorthormi Glacier has proglacial lakes and calvings on both glacier sides. In this situation, it is not easy to define the glacier width (fluxgate width) and the bedrock topography (ice thickness). We considered that it would be rather less robust of ice emergence to estimate glacier width from only remote-sensing data. Therefore, we used the assumption of a fixed glacier width and thickness.

We also understand the uncertainty of modeled ice thickness. However, other studies also use glacier ice thickness /bed geometry estimated from the inverse modeling dataset as "consensus ice thickness" (e.g., Watson et al., 2020; Wei et al., 2021) without modification. In this study, we used ice thickness estimated in Tsutaki et al. (2019), and they validated and tuned the model (Farinotti et al., 2009) with observed surface velocity and measured ice thickness (terminus lake depth). Therefore, the ice thickness we used is not "actual ice thickness" but is considered to be better constructed than consensus ice thickness.

**L170:** You are using velocities along the centreline yet Eq. 2b works for depth- and width- averaged velocities. There may be an important change in lateral drag due to the presence of lakes on the sides of the Thorthormi glacier, thus the cross-sectional distribution of the ice flow is not constant along the flowline.

We thank you for your useful comment. Regarding the simplified assumption (calculating on center flowline), please see the reply to lines 169-170. We understand that the traverse velocity is not constant; however, the active icebergs separation at both sides of the glacier makes it substantially difficult to calculate the surface velocity (please see the map of the velocity field in general comment). For these reasons, we did not use the transverse velocity.

The presence of outer ice flux on both sides does not significantly affect the discussion because it is a factor that reduces ice emergence (compression). We had mentioned it in lines 295-298 as "We simply assumed a constant glacier width to calculate $V_e$ along the central flowline. However, this glacier terrain tends to widen in the downglacier direction, yielding an extensional velocity regime. The lateral proglacial lakes on both sides of the terminus before it transitioned to a lake-terminating condition may have further contributed to a more negative $V_e$ than that estimated along the central flowline."

**L172:** Farinotti, not Farinottie

We will correct it. Thank you.

**L174:** Here you are assuming no basal motion whereas one can expect an increasing sliding towards the glacier terminus in case of a lake-terminating glacier (Tsutaki, 2019). You are incorrectly referring to Sakai et al. (2006) and Berthier and Vincent (2012), here are quotes from those articles:

Sakai et al. 2006:

"The ice velocity averaged over the depth (h) is taken to be 80% of the surface velocity ($u_s$) by assuming that the ice flow is laminar, that there is no basal motion (consistent with the glacier being of the polar type) and that the empirical constant (n) in Glen's law is 3 (Paterson, 1994)."

Berthier and Vincent, 2012:

"The next step is the conversion from mean surface to depth-averaged velocity. Without basal sliding, theoretical calculations suggest that the depth-averaged velocity is 80% of the surface velocity (for n = 3 in Glen's law; Cuffey and Paterson, 2010, p. 310). With basal sliding, this percentage increases, and for example in the case of Athabasca Glacier, Canada, the mean cross-sectional velocity equals the mean surface velocity (Raymond, 1971). Here we used an intermediate value assuming that the depth-averaged velocity is 90% of the surface velocity."

We agree with your comments. Please see the reply to the general comment.

**L230:** Why did you calculate the velocity field only for 2016/2017 season? I would recommend showing a map of the velocity field.

Thank you for the suggestion. Please see the reply to the general comment.

**L249-257** A plot of the emergence velocity is strongly recommended.

We will add a plot of the emergence velocity. Please see the general comment.

**L250:** The ± sign is not needed as σ is always positive. What is the reason for such a high standard deviation?

We will correct it. Thorthormi Glacier has a large variation in surface flow velocity. Therefore, the standard deviation in emergence velocity is also large. Please see the figure of emergence velocity in the general comment.

**L277:** What is the frontal ablation flux of both glaciers? I think it is important to include it in your analysis as this is the term that closes the mass budget, otherwise, how would you explain the increased thinning and acceleration of ice flow during land-terminating to lake-terminating transition? Please refer to the relevant discussion made by Pronk et al. (2021). There is an abundance of icebergs in the Thorthormi lake as shown in Fig. 1, suggesting a significant frontal ablation flux.

We thank you for your useful comments. Please see the reply to the general comment.

**References:** There are many errors, missing pagination or volume numbers. Please update references that changed from early access to the final publication stage.

We will check all references carefully again and correct errors. Thank you for mentioning it.

**Figure 1:** Please add a description of the black bounding box in panel (b). As far as I'm concerned, the panel labels should be in brackets: (a), not a)

We will add a description that indicates "Panel (a)" for the black bounding box in panel (b). We had already mentioned it in Figure 1 caption as "The dashed box in (b) shows the domain of (a).". We will also change all panel labels to like "(a)" from "a)." Please see revised Figure 1b on the reply to the general comment.

**Figure 2:** This Figure is very similar to Fig. 4 in Tsutaki et al (2019), I guess the reason is to make them easy to compare, please add a reference.

We will add a reference in Figure 2 caption as " The 2000–2012 lake outlines are from Tsutaki et al. (2019)."

The glacier area change increases already 3-4 years before the detachment from the moraine (panel c), this is probably due to the development of lateral lakes - please clarify this in the text

We agree with this comment. We will change line 277 to explain it as " Lateral lakes had formed on both sides of the Thorthormi Glacier terminus several years before 2011 (Fig 2a and 2c)."

**Figure 3:** There is a large scatter in 2011-2018 data, the points diverge into two populations - a positive and a negative one, why?

Please see the reply to line 116 comment. We will add dh/dt plot that is calculated from resampled DEMs.

**Figure 4:** Please add the uncertainties of 2016/17 and 2017 (ITS_LIVE) data series.

We added more ITS_LIVE data to Figure 4; therefore, drawing error bars with all lines would complicate the figure. Hence we will mention ITS_LIVE uncertainty in the method section (Sect.3.3) as " We extracted the annual velocity data along the central glacier flowlines from the ITS_LIVE product (2010–2018) to trace the temporal changes in flow velocity; the ITS_LIVE velocities possess mean uncertainties of 1.0 m $a^{-1}$ (maximum of 3.1 m $a^{-1}$) and 0.6 m $a^{-1}$ (maximum of 6.0 m $a^{-1}$) along Thorthormi and Lugge glaciers, respectively.". We had mentioned the uncertainty of 2016/17 in the same section.

**Additional references:**

[revised manuscript text omitted]